# Quality of care in sterilization services at the public health facilities in India: A multilevel analysis

Vinod Joseph. K. J.[1], Arupendra Mozumdar[2]*, Hemkhothang Lhungdim[1‡], Rajib Acharya[2‡]

1 Department of Public Health and Mortality Studies, International Institute for Population Sciences, Mumbai, India, 2 Reproductive Health Division, Population Council, New Delhi, India

☯ These authors contributed equally to this work.
‡ These authors also contributed equally to this work. HL and RA are joint senior authors.
* amozumdar@popcouncil.org

**Data Availability Statement:** Data of NFHS are available in the public domain without any identifiers in the following website—https://dhsprogram.com/data/available-datasets.cfm Data of DLHS are available in the public domain in the

## Abstract

Female sterilization is the most popular contraceptive method among Indian couples, and the public sector is the major source of sterilization services in the country. However, concerns remain on the quality of services provided, deaths, failures, and complications following sterilization. In this paper, we study the complexities around the quality of care in female sterilization services at public health facilities and identify strategies for improving the measurement of such quality. A better understanding of these issues could inform pragmatic strategies for enhancing quality. This study uses data from the National Family Health Survey (NFHS) 2015–16 and District Level Household and Facility Survey (DLHS) 2012–13. The study is limited to only districts whose data are available in both DLHS 2012–13 and NFHS 2015–16. The methods of analysis include bivariate statistics, Pearson's chi-square test, and two-level mixed-effects logistic regression. We found that the quality of care (QoC) in sterilization service at the public health facilities in India is associated with facility readiness and the socio-economic characteristics of the clients. There is a significant association between household wealth and the QoC received. Our study provides empirical shreds of evidence on the role of structural attributes in delivering quality sterilization services. The spatial analyses revealed the geographies in the country where the QoC and facility readiness are low. Quality should be an overriding priority to establish the credibility of any health care delivery system. It is essential to provide safeguards against adverse events to develop the client's confidence in the services, which is the key to success for any voluntary family planning program like in India.

## Introduction

Quality of care (QoC) in family planning services is not only recognized as a key to improving the health of women and children but also as a human right [1]. All individuals have an equal right to access, choice, and the benefits of advancements in the selection of family planning

following website— http://rchiips.org/DLHS-4.
html.

**Funding:** This paper was prepared as part of a
mentorship program under the RASTA initiative of
the Evidence Project of the Population Council. The
Evidence Project is made possible by the generous
support of the American people through the United
States Agency for International Development
(USAID) under the terms of cooperative agreement
no. AID-OAA-A-13-00087. The contents of this
manuscript are the sole responsibility of the
authors and do not necessarily reflect the views of
USAID or the United States Government. The
funders had no role in study design, data collection
and analysis, decision to publish, or preparation of
the manuscript.

**Competing interests:** The authors have declared
that no competing interests exist.

methods. The World Health Organization defines the QoC as "the extent to which health care services provided to individuals and patient populations improve desired health outcomes. In order to achieve this, health care must be safe, effective, timely, efficient, equitable, and people-centered" [2]. Elements of the QoC in family planning include the choice among a wide range of contraceptive methods, information on the effectiveness, risks and benefits of different methods, and safeguarding privacy and confidentiality.

The demographic concerns drove the early phases of family planning initiatives across the globe, and the objective was to control the rapid population growth and reduce fertility. This leads to a stronger emphasis on quantitative aspects over quality and has predominated among international family planning. Another possible reason is the difficulty in measuring quality while it is comparatively easier to estimate the number of contraceptives distributed or the number of sterilizations done. Over the last decades, the QoC has become a central focus of service providers in the efforts to improve the delivery of family planning services [3]. Later even those populations driven by demographic goals and program targets started to recognize the importance of quality in family planning programs and the linkages between quantity and quality. And there were many supporters to the quality movement on the humanistic premise arguing women deserve to receive the best services possible within the limits of local [4–6].

Donabedian developed the classic framework on measuring the QoC that makes a distinction between structure, process, and outcomes [7–9]. The Institute of Medicine [10] suggests that the efforts to improve care quality should be focused on effectiveness, efficiency, equity, patient-centeredness, safety, and timeliness. Bruce [11] argued attention to a neglected quality dimension of family planning. She introduced a framework, also known as the Bruce-Jain framework, for assessing quality from the client's perspective including choice of methods, the information given to clients, technical competence, interpersonal relations, follow-up and continuity mechanisms, and the appropriate constellation of services. The modified QoC framework [12] includes the choice of contraceptive methods, the information given to users, provider competence, client/provider interactions, re-contact, and follow-up mechanisms, and the appropriate constellation of services.

India is the first country to adopt a national family planning program at the beginning of the 1950s to address the issues of high fertility and rapid population growth [13]. The family planning program in India has evolved and currently not only targets to achieve population stabilization but also serves as a medium of intervention for promoting reproductive and child health. It focuses on access and assuring complete knowledge to reproductive services and enables citizens to make an individual reproductive choice. The program is designed in-line with the goals and objectives of various policies Such as National Population Policy 2000, National Health Policy 2002, and National Health Mission and compliments India's commitment at international forums like Family Planning 2020 [14].

Female sterilization remains the most popular contraceptive method among Indian couples, even though the target-oriented, method-specific approach was stopped in 1996 [15–17]. According to NFHS 2015–16, the overall contraceptive prevalence rate (CPR) in the country is 54% out of that 36% use female sterilization, followed by male condoms (6%), pills (4%), and six percent use a traditional method, mostly the rhythm method [18]. There is a slight decline in the percentage of female sterilization adopters in the past decade from 37% in 2005–06, compared with 34% in 1998–1999 and 27% in 1992–1993 [19] while the percentage of couples relying on male sterilization remains substantially lower and declined to 0.3% in 2015–16 from 4% in 2005–06 compared with 2% in 1998–1999 and 4% in 1992–1993. In India, two female sterilization methods are available. The first one is mini-laparotomy involves making a small incision in the abdomen and the fallopian tubes are brought to the incision to be cut or blocked. The second one is laparoscopy involves inserting a long thin tube with a lens in it into

the abdomen through a small incision and block or cut the fallopian tubes. Both procedures can be done only by a trained and certified gynecologist or surgeon [14].

The public health sector is the major source of sterilization services and IUDs/PPIUDs [20], whereas the private health sector is the major source for injectable, pills, and condoms. Almost 82% of the women, who got sterilized, underwent the procedure in a public health facility, and out of that 53% were done in government/district/municipality hospital (GH/DH/MH), 24% in Community Health Center's (CHC), 13% in Primary Health Center's (PHC), and 10% in mobile clinics and camps organized by the public health system. As per the recent statistic of the Ministry of Health and Family Welfare, a total of sterilizations 3,363,009 were performed in the financial year of 2018–19 [14].

Despite the wide acceptance of sterilization, it is observed that the services provided currently in the country doesn't meet the needs of the people due to various factors, such as insufficient availability of service centers, absence of skilled providers, etc. [21]. There is a continuing concern about the deaths, failures, and complications following sterilization in the country [22]. Available literature mostly considers aspects of the availability of human resources, healthcare infrastructure, services provided, equipment and supplies, coverage, and outcomes [23, 24]. The QoC does not act in isolation, but governance, management, social, community, and structural factors also determine the performance of a health care system [25]. Donabedian model shows that structural factors affect process measures, which in turn affect outcome measures. Structure measures reflect the attributes of material resources like infrastructure, availability of drugs and equipment, and human resources such as availability of an adequate number of personnel, who have requisite knowledge and skills [26]. Previous studies have shown that there exist inequalities in health care delivery and differentials in the QoC based on wealth, ethnicity, socioeconomic status, age, marital status, family status, sex, and disability which put vulnerable sections at a disadvantage [27, 28]. Many studies have identified that lack of structural facilities is a major impediment to public health facilities in providing better health care services in India [29–31].

The family planning services in India and its quality has been extensively documented and critiqued over the last several decades due to its demographic importance in influencing the population dynamics of the country. India still falls far behind in providing QoC in family planning to its citizens, especially in sterilization services. In this paper, we study the complexities around the QoC in female sterilization services at public health facilities and identify strategies for improving the measurement of such quality. A better understanding of these issues could inform pragmatic strategies for improving the QoC.

## Methodology

### Theoretical framework

The literature shows, a number of theoretical frameworks have been proposed to study the QoC for family planning services. For this study, we have adopted the framework proposed by Jain et al. [32]. The framework has three major QoC components: structure—denoting the infrastructure and readiness of the system and the facilities delivering FP services, process—denoting the delivery of the services, and outcome—denoting the consequences of the service delivery. The outcome could have negative consequences such as infection, complication, death, method failure, etc. or could have positive consequences such as client satisfaction, and method continuation.

The client-provider interactions are the centerpiece of family planning programs, and it identifies appropriate ways to obtain information on how the service-delivery systems interact with people they are intended to serve [33]. It is crucial for improving the quality of family

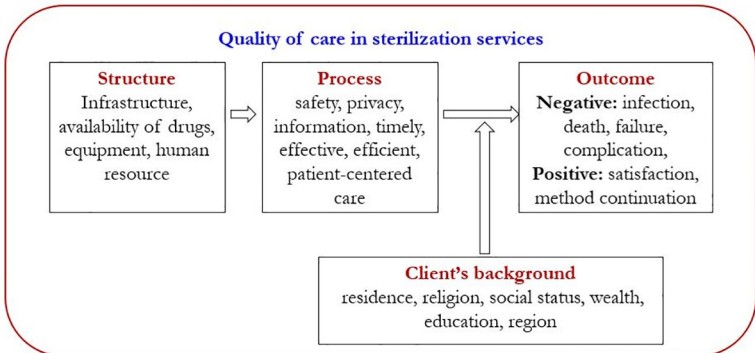

**Fig 1. Framework of quality of care in sterilization services.**

planning services and the most important independent predictor of informed choice among family planning clients [34, 35]. However, other aspects of service delivery, such as access, availability, and the constellation of methods provided are also vital aspects of quality [36].

For this study, we theorized that the process-outcome also depends on the background characteristics of the client. The client-provider interaction being a significant component of the process quality, often depends on the background characteristics of the clients. The clients from the marginalized communities often do not get enough information about making an informed choice [37]. Also, the reporting of the process quality and outcome depends on the client's perception and expectation on quality. Therefore, in the theoretical framework for this study, we considered the client's background characteristics as a major component that affects the receipt and the reporting of the QoC in family planning services (Fig 1).

This study will examine the followings—(1) the association of receipt of QoC with the availability of infrastructure and facility readiness for providing the family planning services and (2) the association of background characteristics of the clients with the receipt of QoC in sterilization services. We hypothesize that the possibility of receipt of quality sterilization services will be less among those women who live in the districts with poor infrastructure and readiness for providing quality family planning services and among the women of marginalized segments of the population.

## Data

The data for this study has been taken from the National Family Health Survey (NFHS) 2015–16 and the District Level Household and Facility Survey 2012–13. The NFHS is a large-scale, multi-round survey conducted on a representative sample of households throughout India. NFHS series provides information on population, health, and nutrition for India and each state and union territory. All four rounds of NFHS have been conducted under the stewardship of the Ministry of Health and Family Welfare (MoHFW), Government of India. MoHFW designated the International Institute for Population Sciences (IIPS), Mumbai, as the nodal agency for all of the surveys [18].

The DLHS 2012–13 is also a nationwide survey that targets to provide reproductive and child health-related information at the district level in India. The data from these surveys have been useful in setting the benchmarks and examining the progress the country has made after the implementation of RCH Programs. The evidence generated by these surveys has also been helpful for the monitoring and evaluation of on-going programs and the planning of suitable strategies by the central and state governments [38].

The information on district hospitals and community health centers from DLHS 2012–13 facility survey is used in this study. The facility data from 80 districts were not provided in the available dataset in the public domain; as a result, this study is limited to only those 560 districts whose data is available in both DLHS 2012–13 facility data and NFHS 2015–16. The specific study area is depicted in Fig 2. Out of 699,686 total women aged 15–49 interviewed in the NFHS 2015–16, 160,242 women who had undergone sterilization in public health facilities

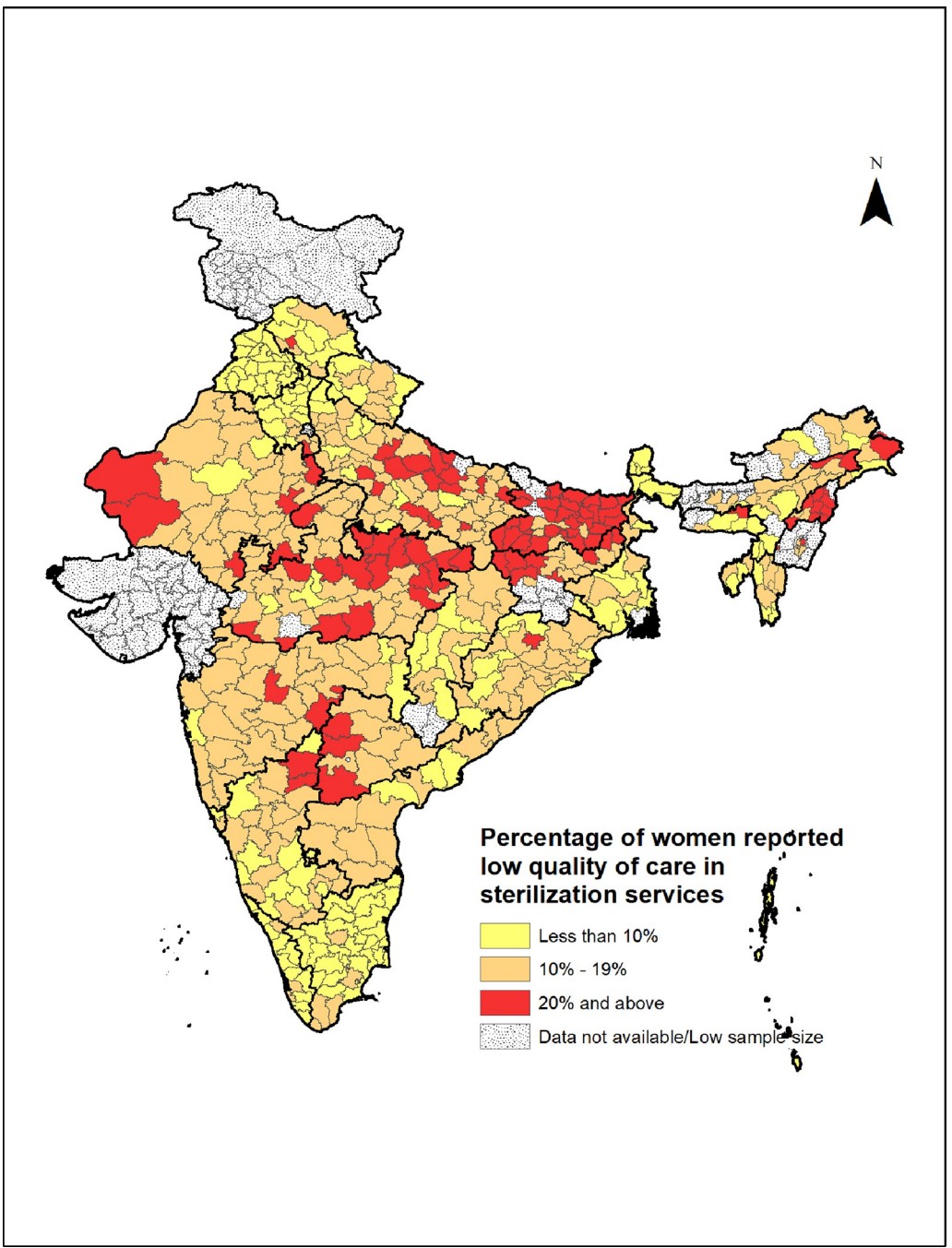

**Fig 2. Distribution of Indian districts by the percentage of clients reported 'low' quality of care score for sterilization service, NFHS 2015–16.**

were identified. And after removing 149,41 women samples from 80 missing districts, 145,301 women from 560 districts were taken as the final sample for this study.

The NFHS 2015–16 and DLHS 2012–13 received ethical clearance from the Ethical Review Board of the International Institute for Population Science, which conducted these surveys. The analyses presented in this study are based on secondary analyses of existing survey data, available in the public domain without any identifying information. The survey personnel obtained informed consent from each respondent before the interview and made their best effort to maintain privacy.

## Variable description

**Dependent variable.**   The outcome variable for this analysis is a process-outcome variable which is an index measurement composed of five variables related to process and outcome-related variables on the QoC of sterilization services. The list of those five variables and the recoding of the variables have been presented in S1 Table. We used principal component analysis to create the index measurement of the process-outcome variable of QoC. The index score was computed based on the first principal component, which explained the largest proportion (36.3%) of the total variance. The index was created with a Cronbach's alpha of 0.71. The index scores ranged from -2.58 to 0.387. The distribution of the index score showed a bimodal distribution with a mean score of 0, SD of 1, a median of 0.39, and skewness of -2.2. Therefore, we took '0' as the cut-off to divide the scores into two groups. The individual with negative scores has been considered receiving 'low' QoC and the individual with a positive score has been considered receiving 'high' QoC. Since from a program implementation viewpoint it is more important to identify the factors related to low QoC, we created an individual-level variable and coded each individual either with '1', if she received 'low' QoC, or with '0' otherwise.

**Independent variables.**   *Structure score*. We considered two types of variables as the independent variables: variables related to infrastructure and the readiness of the facilities and the variables on background characterizes of the clients. To measure the readiness of the facilities to provide FP service we used a set of variables measuring material resources like infrastructure, availability of drugs and equipment, and availability of the trained personnel, etc. and summarized those in a 'structure score'. The score was calculated from DLHS 2012–13 facility survey data—based on thirty-two indicators. The survey provides information about public health facilities such as district hospitals, community health centers, primary health, and sub-health centers at district levels. We only used information about DH and CHC's because all DH and CHCs within the district were covered in the facility survey, and no sampling procedure is involved [38]. Also, we assumed that the readiness of the DH and CHC would provide a proxy for the overall readiness of the health system to provide sterilization services. A detailed description of these variables is provided in S2 Table.

The structure scores were derived using the principal component analysis based on the first principal component, which explained the largest proportion (16.3%) of the total variance. The index was created with a Cronbach's alpha of 0.81. The index scores ranged from -3.151 to .3173. The score showed a negatively skewed distribution with a mean score of 0.003, SD of 0.45, a median of 0.167, and skewness of -2.15. For each district, we calculated the mean structure score of all the facilities (district hospitals, sub-district hospitals, and CHCs) in that district. Then we considered the district with a negative mean score having a 'low' structure score, which denotes that the district having poor infrastructure and readiness, and the district with a positive mean score having a 'high' structure score. Therefore, we created a binary variable at the district level and coded each district either '0' if having a low structure score or '1' if having a 'high' structure score.

The district-level variable was merged with the individual-level dataset such that all women in a district had the same value of the variable denoting the structure score. The DLHS 2012–13 facility survey does not provide data from all Indian districts. Therefore, we have been able to merge the data form 560 out of 640 Indian districts.

*Background characteristics of the women*. A set of background characteristics of the women was considered in this analysis. These variables were found to be associated with the receipt of the QoC for sterilization services in earlier studies [27–31]. The list of such variables and the recoded categories are presented in S3 Table.

## Statistical analysis

The socio-demographic and facility-level characteristics were described using descriptive statistics. Pearson's chi-square tests were used to compare the differences in receipt of QoC across women of different background characteristics.

As mentioned earlier, we prepared a hierarchal data structure, i.e. the women are nested within the districts. Therefore, the use of conventional regression models could underestimate standard errors of the effect sizes, which consequently can affect the decision for testing the hypotheses. Since we hypothesized that the clients within the same district might get the chance of a similar QoC in sterilization services compared to those clients from other districts, the analysis involving only one level of data violates the assumptions of equal variance and independence of observations. Thus, we used a two-level mixed-effects logistic regression model to test the effect sizes of individual and district-level factors of QoC in sterilization services. Mixed-effects logistic regression is logistic regression containing both random effects and fixed effects. The conditional distribution of the response given the random effects is assumed to be Bernoulli, with success probability determined by the logistic cumulative distribution function [39]. It allows for not just one, but many levels of nested clusters of random effects. The respondents comprise the first level, and the districts comprise the second level. The model specifies the random effects for districts.

Evidence shows that different levels of geographical areas such as clusters, districts, states, and regions have a disparate impact on the behavior of the respondents within them, owing to the variability in social-economic characteristics as well as differences in family planning programs and policies. Therefore, it is imperative to explore levels of heterogeneity between districts and regions.

We opted for running four models using the receipt of QoC for sterilization service as dependent variable—if a woman received 'low' QoC we coded her with '1', otherwise with '0'. Model-1 is empty, which was fitted without explanatory variables to test random variability in the intercept. Model-2 examined the effects of health facility-level characteristics. Model-3 examined the effect of socio-demographic factors, and Model-4 examined the effects of both health facility and socio-demographic characteristics simultaneously. The mixed-effects logistic regression function can be written as:

$$Log\left(\frac{\pi_{ij}}{1-\pi_{ij}}\right) = \beta_0 + \beta_1 X_{1ij} + \cdots + \beta_n X_{nij} + u_{oj} + e_{ij} \tag{1}$$

Where, $\pi_{ij}$ is the probability of low QoC in sterilization services, $\beta_0$ is the log odds of the intercept, $\beta_1 \ldots \beta_n$ are effect sizes of health facility and socio-demographic factors, $X_{1ij} \ldots X_{nij}$ are independent variables of health facility and socio-demographic characteristics and $u_{oj}$ and $e_{ij}$ are random errors at district and individual levels [39–41].

In the multilevel models, the fixed effects estimate the association between the likelihood of QoC in sterilization services, and the individual and district-level factors are expressed as odds

ratios with 95% confidence intervals. The random-effects estimate the variation in the QoC across different groups expressed as Intra-Class Correlation (ICC) and Proportional Change in Variance (PCV).

The ICC estimates the correlation or the dependence between latent linear responses conditional on the fixed-effects covariates for different levels of nesting [42]. The ICC is calculated as:

$$ICC = \frac{\sigma u^2}{\sigma u^2 + \frac{\pi^2}{3}} \qquad (2)$$

Where, $\sigma u^2$ is the variance at the district level and $\frac{\pi^2}{3}$ is the variance at the individual level [43–45]. ICC shows the level of between-district correlation within a model and compares the successive models by observing at the decline in ICC.

The PCV estimates the variability on the odds of QoC in sterilization services explained by the successive models to the empty model. The PCV is calculated as:

$$PCV = \frac{V_e - V_{mi}}{V_e} \qquad (3)$$

Where, $V_e$ is the variance in the empty model and $V_{mi}$ is the variance in successive models [41].

The models are diagnosed using the Log-likelihood Ratio test (LR test) and the Akaike's Information Criterion (AIC). An LR test compares the fitted model to ordinary logistic regression and the p-value <0.05 shows the statistical significance of the fit and the adequacy of using the multi-level model over a conventional logistic [46].

AIC calculates the information criteria used to compare models. Akaike's [47] information criterion is defined as:

$$AIC = -2\ln L + 2k \qquad (4)$$

Where $\ln L$ is the maximized log-likelihood of the model and $k$ is the number of parameters estimated. AIC is expected to decline as more variables are added to the successive models. The model with smaller AIC fits the data better and confirms the goodness-of-fit.

In the NFHS the sample weights were calculated for the cluster level (PSUs). Multilevel models require weight variables corresponding to each level such that the weights should be constant within the groups defined. In this study, the respondents comprise the first level, and the districts comprise the second level. As we didn't consider cluster (or PSU) as a level and weights were not constant within the district, we didn't use the sampling weights in the multi-level regression analysis.

All statistical analyses were performed using STATA 15 [48]. All India states shapefiles used in the study were downloaded from the DHS program spatial data repository [49]. The final feature class had 642 polygons representing each survey district in NFHS-4. All the spatial analyses were conducted in ArcGIS 10.5 [50] with 999 permutations and a pseudo p-value for the cluster of <0.05 computed.

## Results

Out of 145,301 clients selected for the study, 27% lived in urban areas, and 73% lived in rural areas (Table 1). Most of the women (87%) were Hindus, six percent were Muslims, two percent were Christians, and around four percent belonged to other religious communities including Sikhs, Jains, and Buddhists. More than half (54%) of the women had undergone sterilization in

**Table 1. Descriptive statistics, percentage distribution, and results of Pearson's chi-square test for the quality of care in sterilization services in India.**

| Background characteristics | Sample | Column % | Percentage of women received 'Low' quality of care | Chi-square | p-value |
|---|---|---|---|---|---|
| **Residence** | | | | | |
| Urban | 39224 | 27.0 | 9.19 | 479.17 | p< 0.001 |
| Rural | 106077 | 73.0 | 13.93 | | |
| **Religion** | | | | | |
| Hindu | 127255 | 87.6 | 12.83 | 233.36 | p< 0.001 |
| Muslim | 9430 | 6.5 | 13.03 | | |
| Christian | 3320 | 2.3 | 11.09 | | |
| Others | 5296 | 3.6 | 8.63 | | |
| **Social group[1]** | | | | | |
| Scheduled caste | 35231 | 24.3 | 13.26 | 326.67 | p< 0.001 |
| Scheduled tribe | 14791 | 10.2 | 16.51 | | |
| OBC | 64687 | 44.5 | 12.35 | | |
| Others | 30592 | 21.1 | 10.73 | | |
| **Wealth quintile** | | | | | |
| Poorest | 25165 | 17.3 | 21.39 | 3100.00 | p< 0.001 |
| Poorer | 32142 | 22.1 | 14.42 | | |
| Middle | 35892 | 24.7 | 11.67 | | |
| Richer | 31848 | 21.9 | 9.24 | | |
| Richest | 20255 | 13.9 | 6.11 | | |
| **Education** | | | | | |
| No education | 64707 | 44.5 | 11.10 | 153.71 | p< 0.001 |
| Primary education | 26201 | 18.0 | 12.41 | | |
| Secondary and above | 54393 | 37.4 | 14.62 | | |
| **Region** | | | | | |
| North | 20572 | 14.2 | 10.60 | 916.84 | p< 0.001 |
| Central | 29457 | 20.3 | 14.81 | | |
| East | 25116 | 17.3 | 15.90 | | |
| Northeast | 1310 | 0.9 | 13.59 | | |
| West | 19446 | 13.4 | 14.20 | | |
| South | 49401 | 34.0 | 9.94 | | |
| **Place of sterilization** | | | | | |
| GH/DH/MH | 77742 | 53.5 | 10.69 | 540.44 | p< 0.001 |
| CHC | 34832 | 24.0 | 15.23 | | |
| PHC/Sub-center | 18552 | 12.8 | 15.62 | | |
| Camp/Mobile clinic/Others | 14175 | 9.8 | 13.20 | | |

[1] Scheduled caste, scheduled tribes, and other backward classes (OBCs) are marginalized groups in India designated by the government and recognized by the Constitution of India.

the district or municipal hospitals (GH/DH/MH), one-fourth in community health centers, 13% in primary health centers, and 10% percent in sterilization camps.

The percentage of women who received each item of QoC of sterilization services showed that the highest percentage of clients rated the service as 'good' or 'all right' (95.5%) (S1 Table) and most of the item of QoC was received by around 80–90% of women. Overall, 12.7% of the sterilization clients receipt the services with 'low' quality, i.e. that proportion of clients reported negative scores in overall process-outcome measures. A variation in the receipt of QoC for sterilization services can be found as more than 10% of the clients from 389 districts, and more

than 20% of the clients from 100 districts reported receiving 'low' QoC scores (Fig 2). The percentage of clients who received 'low' quality services were significantly higher for the client from rural areas, belonged to Muslim, scheduled caste, and scheduled tribe communities, poor households, from eastern and central regions of India, and those who received sterilization from CHCs, PHCs, or sub-centers (Table 1).

Overall, the availability of infrastructure and facility readiness in districts of India are 'poor' (S2 Table). The percentage of facilities which had the readiness for human resources (mostly around 60–70%) is the highest and most of the item regarding infrastructure were not available in the facilities. Overall, the CHCs showed better readiness than the district hospitals in terms of availability of infrastructure and human resources. Out of 560 districts covered, 200 districts had a 'low' structure score for infrastructure and readiness. About 16% of the clients who were from the district with 'low' structure score also reported 'low' QoC score, whereas 12% of the clients who were from the 'high' structure score reported so and the Chi-square test showed a significant association of structure score and QoC score.

The geographic distribution of the districts with 'low' structure score showed that a large number of districts from the northern and eastern part of the country, which is also similar to the geographic distribution of the higher percentage of receipt of the 'low' QoC score (Fig 3).

The State/Union Territory (UT)-wise means of the QoC in sterilization services at the public health facilities shows that Kerala, Chandigarh, Goa are the top three high performing states and Jharkhand, Bihar, Odisha are the bottom three low performing states/UT's in providing good QoC in sterilization at the public health facilities (S1 Fig). The State/UT-wise means of structure score, indicating the infrastructure available at the public health facilities shows that Kerala, Goa, and Puducherry are the top 3 high performing states and Bihar, Jharkhand, Sikkim are the bottom three low performing states in providing good structural facilities at the public health facilities (S1 Fig).

## Multilevel analyses

The results of the multilevel mixed-effects logistic regression are shown in Table 2. Model-1 (empty model) shows statistically significant variability in the odds of receiving the excellent QoC in sterilization services (district-level variance: 0.4011, p<0.001), and the 11% of the total variance is accounted by between-district variation of characteristics (ICC: 0.109). In Model-2, the health facility variables were added. The results show that the type of health facility and infrastructure is significantly associated with the quality of the care provided. The ICC in Model-2 indicated that 10% of the variation in the QoC was attributable to differences across districts. The PCVs show that the health facility level characteristics explained 11.64% of the variance across communities. In Model-3, the socio-demographic variables were included. The results depict that the place of residence, household wealth, and education are significantly associated with the QoC. The ICC in Model-3 implied that differences between districts account for about 8% of the variation in the QoC in sterilization services and PCV indicates that socio-demographic characteristics explain 30.61% of the variation between districts.

In Model-4, both health facility level and socio-demographic level variables were included simultaneously. As shown by the estimated ICC, 7.5% of the variability in the QoC in sterilization services is attributable to differences between districts. The PCV indicated that 31% of the variation is explained by both health facility and socio-demographic factors included in Model-4.

The fixed effects of the Model-4 show that the odds of receiving low-quality care in sterilization services among rural women is 1.08 times (Adjusted Odds Ratio or AOR: 1.079 [1.025–1.136]) higher than urban women. When the religious affiliation is considered, compared to

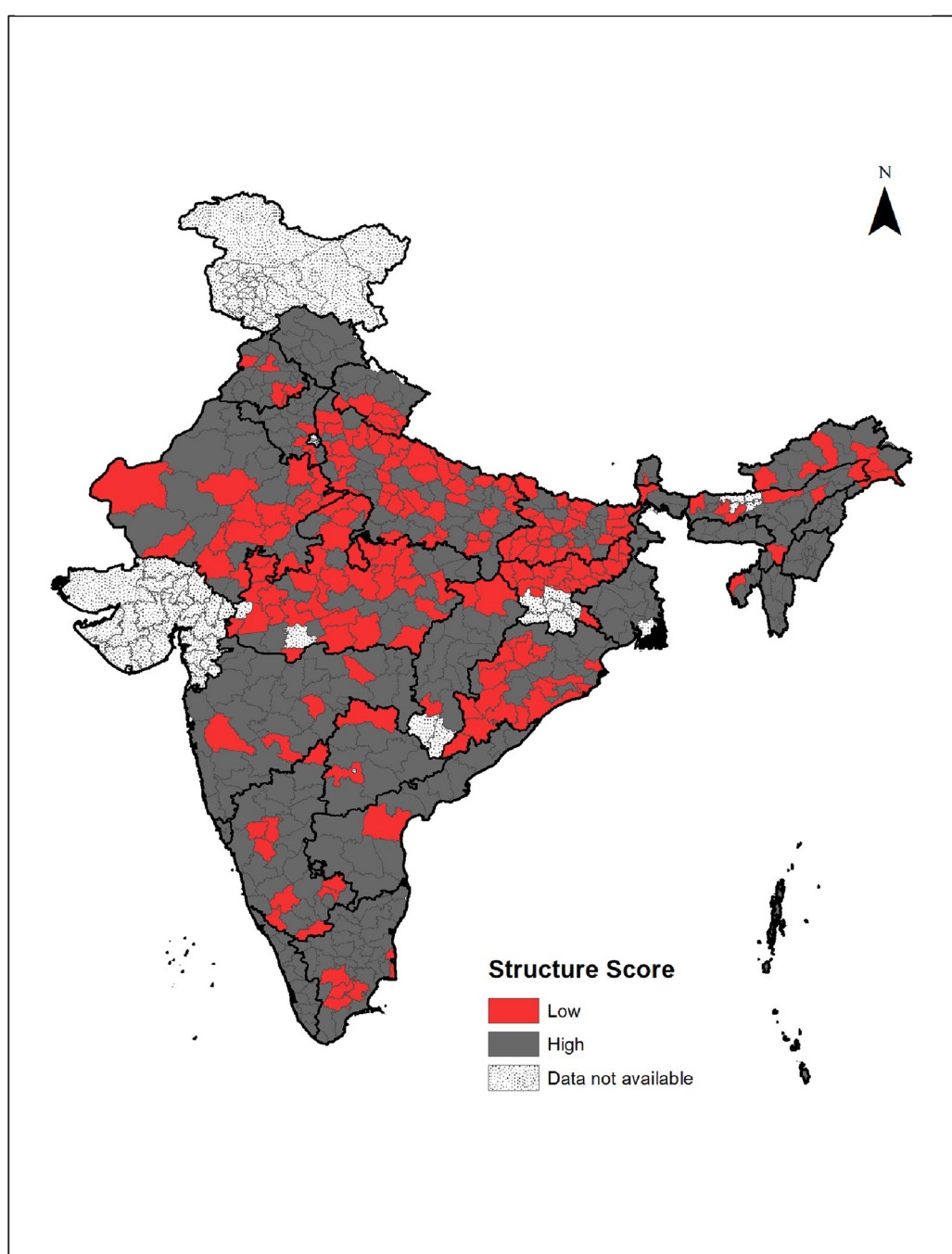

**Fig 3. District-wise distribution of 'low' structure score indicating the availability of infrastructure and readiness at the public health facilities in India, DLHS facility survey 2012–13.**

Hindu women the Muslim women and Christian women have 1.45 times (AOR: 1.447 [1.336–1.568]) and 1.13 times (AOR: 1.130 [0.990–1.290]) higher odds of receiving low-quality sterilization services respectively. On comparing the social status, the scheduled caste's, scheduled tribes, and OBC's have 1.13 times (AOR: 1.134 [1.084–1.187], 1.2 times (AOR: 1.199 [1.132–1.270]) and 1.18 times (AOR: 1.180 [1.119–1.244]) higher odds of receiving low-quality sterilization services compared to other social groups.

**Table 2. Multilevel logistic regression analysis of factors associated with 'low' quality of care in sterilization services in India.**

| Background characteristics | Model-1 | Model-2 AOR (95% CI) | Mode-3 AOR (95% CI) | Model-4 AOR (95% CI) |
|---|---|---|---|---|
| **Individual-level variables** | | | | |
| **Residence** | | | | |
| Urban | - | - | 1.00 | 1.00 |
| Rural | | | 1.085** (1.031–1.141) | 1.079** (1.025–1.136) |
| **Religion** | | | | |
| Hindu | - | - | 1.00 | 1.00 |
| Muslim | | | 1.449** (1.338–1.569) | 1.447** 1.336–1.568) |
| Christian | | | 1.117 (0.978–1.276) | 1.130 (0.990–1.290) |
| Others | | | 0.716** (0.631–0.811) | 0.716** (0.632–0.812) |
| **Social groups** | | | | |
| Scheduled caste | - | - | 1.133** (1.083–1.187) | 1.134** (1.084–1.187) |
| Scheduled tribe | | | 1.198* (1.131–1.269) | 1.199** (1.132–1.270) |
| OBC | | | 01.182** (1.121–1.246) | 1.180** (1.119–1.244) |
| Others | | | 1.00 | 1.00 |
| **Wealth quintile** | | | | |
| Poorest | - | - | 5.630** (5.160–6.142) | 5.609** (5.140–6.120) |
| Poorer | | | 3.395** (3.128–3.685) | 3.392** (3.124–3.682) |
| Middle | | | 2.409** (2.226–2.607) | 2.409** (2.226–2.608) |
| Richer | | | 1.658** (1.580–1.740) | 1.678** (1.551–1.815) |
| Richest | | | 1.00 | 1.00 |
| **Education** | | | | |
| No education | - | - | 1.00 | 1.00 |
| Primary education | | | 1.670** (1.590–1.753) | 1.670** (1.591–1.753) |
| Secondary and above | | | 2.809** (2.687–2.936) | 2.810** (2.688–2.937) |
| **Region** | | | | |
| North | - | - | 1.387** (1.181–1.629) | 1.377** (1.173–1.615) |
| Central | | | 1.626** (4.400–1.887) | 1.520** (1.304–1.770) |
| East | | | 1.636** (1.396–1.916) | 1.500** (1.274–1.766) |
| Northeast | | | 1.217** (1.001–1.481) | 1.197 (0.986–1.454) |
| West | | | 1.479** (1.889–1.840) | 1.476** (1.190–1.830) |
| South | | | 1.00 | 1.00 |
| **Place of sterilization** | | | | |
| GH/DH/MH | - | 1.00 | - | 1.00 |
| CHC | | 1.186** (1.135–1.238) | | 1.070** (1.023–1.119) |
| PHC/Sub-center | | 1.183** (1.120–1.250) | | 1.060* (1.002–1.122) |
| Camp/Mobile clinic/Others | | 0.965 (0.910–1.024) | | 0.885** (0.833–0.940) |
| **District level variables** | | | | |
| **Structure score** | | | | |
| Low | - | 1.360** (1.215–1.520) | - | 1.196** (1.074–1.332) |
| High | | 1.00 | | 1.00 |
| District-level variance (SE) | 0.4011 (0.2891) | 0.3544 (0.0262) | 0.2783 (0.0211) | 0.2684 (0.0205) |
| Log-likelihood | -48689.507 | -48625.163 | -46770.293 | -46742.091 |
| LR test χ2 | 3460.960 | 2780.480 | 2232.370 | 2103.72 |
| P > χ2 | 0.000 | 0.000 | 0.000 | 0.000 |
| ICC | 0.109 | 0.097 | 0.078 | 0.075 |
| PCV | Reference | 11.64% | 30.61% | 33.08% |

(*Continued*)

**Table 2.** (Continued)

| Background characteristics | Model-1 | Model-2 AOR (95% CI) | Mode-3 AOR (95% CI) | Model-4 AOR (95% CI) |
|---|---|---|---|---|
| AIC | 97383.01 | 97264.33 | 93580.59 | 93534.18 |

** p<0.01

* p<0.05

The odds of receiving low-quality care in sterilization services among the poorest is 5.6 times (AOR: 5.609 [5.140–6.120]) higher than that of the richest groups. The result highlights a significant positive association between the wealth and quality of sterilization care received. The women of the poorer, middle, and richer quintiles have 3.39 times (AOR: 3.392 [3.124–3.682]) 2.4 times (AOR: 2.409 [2.226–2.608]) and 1.68 times (AOR: 1.678 [1.551–1.815]) higher odds of receiving low-quality sterilization care than richest counterparts. The women who have secondary or more education and women with primary education reported receiving 1.67 times (AOR: 1.670 [1.591–1.753]) and 2.8 times (AOR: 2.810 [2.688–2.937]) higher odds of low-quality services than women without any formal education. When the geographical regions where considered, the women in the North, Central, East, Northeast, and West regions have respectively 1.37 times (AOR: 1.377 [1.173–1.615]), 1.52 times (AOR: 1.520*** [1.304–1.770]), 1.5 times (AOR: 1.500 [1.274–1.766]), 1.2 times (AOR: 1.197 [0.986–1.454]) and 1.47 times (AOR: 1.476 [1.190–1.830]) higher odds of receiving low-quality sterilization services than women in South region. The Health facility-level factors show that GH/DH/MH and camp/mobile clinic/others (AOR: 0.885 [0.833–0.940]) provide better sterilization services than the CHC's (AOR: 1.070 [1.023–1.119]) and PHC's (AOR: 1.060 [1.002–1.122]). The results of the structure scores show that the health facilities with low Infrastructure have 1.20 times (AOR: 1.196 [1.074–1.332]) higher odds of providing low-quality sterilization services.

The AIC values for the study were 97383.01, 97264.33, 93580.59, and 93534.18, respectively. The AIC results indicate that the full model (Model-4) with the smallest AIC value fits the data better than the other models. There is a consistent decline in the AIC and log-likelihood values, which suggest that the models were adequately fitted. The LR test at each model confirms the goodness-of-fit of the models. Results in all the models provide evidence that health facilities, as well as socio-demographic characteristics, are part of the QoC in sterilization services in the country.

## Discussion and conclusions

We found that the QoC in sterilization services at the public health facilities in India is associated with both facility readiness and the socio-economic characteristics of the clients. The results show that more than one in every ten women sterilized at the public health facilities has received a 'low' QoC. Though the percentage is small, the absolute numbers are alarmingly high with more than 3.3 million women undergoing sterilization per year in public health facilities.

The quality of care for sterilization service is worse in rural areas. There is a significant association between household wealth and the QoC received. Further in-depth studies are required to estimate the magnitude of persisting wealth-based inequality in the QoC and explain the contribution of different socio-economic predictors to it.

Overall, for all the quality-related items, we measured 10–20% of women reported a lack of quality in sterilization service. In terms of percentages, it seems like most of the women are getting sterilization services of 'good' quality. This generalization might not reveal the true

picture. The study shows that educated women reported more likely to receive a 'low' quality of care at public health facilities. This might be due to higher expectations, and better awareness about the QoC and the entitlements that should be received at the public facilities among the educated women compared to their uneducated counterparts. Therefore, the prevalence of receiving 'low' QoC for sterilization services might be even higher than what has been found in this study.

The district/municipality hospitals and sterilization camps were found to provide better quality services than primary and community health centers in the country. The availability of the items which were included to form the 'structure score' also showed a higher presence in the district or sub-district hospitals than community health centers (S2 Table). This highlights the need for strengthening the health facilities at the bottom level to establish the credibility of the public health care delivery system.

In this study, we found that the less percentage of women reported 'low' QoC for sterilization if they received the service at the camps than the percentage of women reported 'low' quality in the camps or mobile clinics. These results seem to be contradictory to population belief and especially after the tragic incidence of the Bilaspur sterilization mishap [51]. However, if we describe how these camps or mobile clinics are organized, the findings could highlight some plausible explanations. The camps and mobile clinics are generally arranged at the geographically inaccessible areas or in remotely located facilities. These camps are often organized with the makeshift arrangements and often the commodities, supplies, and human resources are provided from the higher facilities like district hospitals. These temporary arrangements are often made for that day or a short-time period only. Therefore, the readiness of the camps or mobile clinics may be better than what is available in primary or community health centers.

Our study provides empirical shreds of evidence on the role of structural attributes in delivering better quality services. The spatial analysis revealed those areas where the QoC and structural facilities are low. Policymakers and public institutions should focus on these areas and identify the location-oriented problems and reasons for the low performance. The results inferred through this study are consistent with the previous studies [27–31].

Quality becomes an overriding priority to establish the credibility of any health care delivery system. It is very much essential to measure how well the client's expectations, as well as providers' technical standards and their adherence to those standards, are being met to ensure better quality services [52]. The quality of client-provider interaction, as perceived by clients, is a key to ensuring informed decision making in the delivery of family planning services [35]. Numerous barriers have kept policymakers, and researchers from taking a closer look at client-provider interactions, and more indicators are necessary for family planning program evaluation [33, 34]. The requirement-oriented training targeting clinical competence, coordination, and integration of care are critical measures for improving the QoC in family planning services.

QoC is a determinant of family planning uptake and continuation [4, 53]. The concept of quality of health care services is subjective and difficult to define and measure [54]. The factors contributing to the gap in access to QoC in family planning services are complex, multidirectional, and diverging significantly across regions, ages, families, and communities. Quality of health services plays a vital role in its acceptance. Poor quality of services leads to the client's dissatisfaction and resulting in under-utilization of services [17, 55]. It is essential to provide them safeguards against adverse events to build the client's confidence. Improving the quality of contraceptive services is a significant element that could enhance the acceptance of services.

The perceptions of QoC are shaped by inter-linkages between community, health-system, and individual factors. And the QoC cannot be understood fully without acknowledging the

social norms, relationships, and values within and across the communities and societies where care is provided. The policies and programs should focus on grass root levels where the vulnerable and socio-economically disadvantaged populations are high. Identifying such needs at the micro-level is essential to formulate adequate health programs, policies, and strategies. It can promote not only the well-being of women and children in general but also helps the country to achieve its commitment to sustainable development goals.

As countries continue to progress towards universal health coverage, developing more standardized replicable, and comparable metrics for measuring the quality of medical care should be prioritized. It is very much essential to measure how well the client's expectations, as well as providers' technical standards, are being met to ensure better quality services. It is necessary to monitor and evaluate sterilization services under pre-established criteria and pursue opportunities for improving services. There is a need for replicable, standardized, and comparable metrics for the quality of medical care in low and middle-income countries [56]. Policies to improve population health mainly focus on the expansion of access to essential health services and often neglect the need to address the QoC [25].

Improving the quality of services provided is a significant element that would enhance the acceptance of services. Monitoring and constant assessment of family planning services are essential for delivering quality services. Health facilities must have an appropriate physical environment including uninterrupted power supply, water, sanitation, hygiene, and waste disposal facilities, which are reliable, safe, and functional. Facilities should be designed, organized, and maintained to allow privacy to the clients and facilitate the provision of quality care. Health departments should also ensure that the facilities have adequate stocks of medicines, supplies, and equipment. Health facilities need motivated and well-trained staff available 24/7 to provide better quality services. It is also essential to provide in-service training to improve the quality and sustain providers' knowledge and competence in providing family planning services.

Clients should be provided with adequate information on different family planning methods. It emphasizes them to select the ideal method that best satisfies their reproductive, personal, and health needs based on a thorough understanding. Effective client-provider interactions can reduce unnecessary anxiety and make sterilization a positive experience, even if complications are experienced. All women undergoing sterilization must receive care that prevents hospital-acquired infections which may lead to deaths, failures, and complications. Every woman undergoing sterilization should have a standardized, complete, and accurate medical record for clinical follow-up and early detection of complications.

This study has some limitations. Firstly, the QoC index, used in this study, is constructed with only five variables due to the unavailability of more quality-related variables in the data. As a result, the index may not be an exact representation of all the quality-related aspects. Also, all respondents might not have equal level awareness on QoC, their rights, and entitlements, therefore, the QoC measured here should be considered in a relative term than in absolute terms. Secondly, in our analysis, we assumed that the women went to the nearby facility in her district. However, she could have gone to facilities in another nearby district which provides better facilities than that of her district. Thirdly, while merging NFHS 2015–16 and DLHS 2012–13 datasets, we assumed that there is not much difference in facility readiness in the two survey periods. Due to this assumption, the changes in structural factors during this period are unaddressed. On the other hand, many of the women, who adopted sterilization, underwent the procedure well before the survey year, and the DLHS 2012–13 data for 'structure score' serves as the proxy measurement for the system and facility readiness, which is a precondition for having quality service. Nevertheless, the upcoming rounds of family health surveys should

focus on the facility and quality-related aspects of the health care delivery system as well to understand the measures with full context.

Understanding the nature of the QoC is essential in escalating the priority of health interventions. Perceptions of poor quality of health care may dissuade patients from using the available services. Ensuring the quality of the services will result in improved outcomes at the facility level. To increase the uptake of health services, we require not only medical innovations but also acceptability and a patient-centric approach across the continuum of care. Policies to improve population health should focus on quality rather than only focusing on the expansion of access to essential health services. A quality-oriented approach helps in identifying the gaps in service delivery and tracing its roots and linking them to organizational processes. It builds a system of taking necessary actions for traversing the gaps, periodic assessment, and improving the quality. Providing safeguards against adverse events is essential to develop the clients' confidence in the services, which is the key to the success of a voluntary family planning program like in India.

## Supporting information

**S1 Table. Variables used to create the quality of sterilization care index and percentage of women received each of the quality of care items during sterilization, NFHS 2015–16.** (DOCX)

**S2 Table. Variables used to create 'structure' index and percentage of facilities not having each of the items on infrastructure and readiness by the type of facilities, DLHS facility survey 2012–13.** (DOCX)

**S3 Table. Background characteristics of the clients.** (DOCX)

**S1 Fig. Ordering of the states by mean scores of quality of care and structure scores.** (TIF)

## Author Contributions

**Conceptualization:** Hemkhothang Lhungdim, Rajib Acharya.

**Data curation:** Vinod Joseph. K. J.

**Formal analysis:** Vinod Joseph. K. J., Arupendra Mozumdar.

**Funding acquisition:** Rajib Acharya.

**Investigation:** Vinod Joseph. K. J., Arupendra Mozumdar.

**Methodology:** Arupendra Mozumdar, Hemkhothang Lhungdim.

**Project administration:** Rajib Acharya.

**Resources:** Vinod Joseph. K. J., Arupendra Mozumdar, Rajib Acharya.

**Supervision:** Arupendra Mozumdar, Hemkhothang Lhungdim, Rajib Acharya.

**Validation:** Arupendra Mozumdar.

**Visualization:** Vinod Joseph. K. J., Arupendra Mozumdar.

**Writing – original draft:** Vinod Joseph. K. J., Arupendra Mozumdar.

**Writing – review & editing:** Arupendra Mozumdar, Hemkhothang Lhungdim.

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
