## [Decision Letter · Decision Letter 0]

12 Aug 2020

PONE-D-20-22065

Quality of care in sterilization services at the public health facilities in India: A multilevel analysis

PLOS ONE

Dear Dr. Mozumdar,

Thank you for submitting your manuscript to PLOS ONE. After careful consideration, we feel that it has merit but does not fully meet PLOS ONE’s publication criteria as it currently stands. Therefore, we invite you to submit a revised version of the manuscript that addresses the points raised during the review process.

We look forward to receiving your revised manuscript.

Kind regards,

Kannan Navaneetham, PhD

Academic Editor

PLOS ONE

Journal Requirements:

4. Please ensure that you refer to Figure 1 in your text as, if accepted, production will need this reference to link the reader to the figure.

5. We note that Figures 2 and 3 in your submission contain map images which may be copyrighted. All PLOS content is published under the Creative Commons Attribution License (CC BY 4.0), which means that the manuscript, images, and Supporting Information files will be freely available online, and any third party is permitted to access, download, copy, distribute, and use these materials in any way, even commercially, with proper attribution. For these reasons, we cannot publish previously copyrighted maps or satellite images created using proprietary data, such as Google software (Google Maps, Street View, and Earth). For more information, see our copyright guidelines: http://journals.plos.org/plosone/s/licenses-and-copyright.

5. 1.    You may seek permission from the original copyright holder of Figures 2 and 3 to publish the content specifically under the CC BY 4.0 license. 

5.2.    If you are unable to obtain permission from the original copyright holder to publish these figures under the CC BY 4.0 license or if the copyright holder’s requirements are incompatible with the CC BY 4.0 license, please either i) remove the figure or ii) supply a replacement figure that complies with the CC BY 4.0 license. Please check copyright information on all replacement figures and update the figure caption with source information. If applicable, please specify in the figure caption text when a figure is similar but not identical to the original image and is therefore for illustrative purposes only.

Reviewers' comments:

Reviewer's Responses to Questions

**Comments to the Author**

1. Is the manuscript technically sound, and do the data support the conclusions?

Reviewer #1: Partly

Reviewer #2: Yes

2. Has the statistical analysis been performed appropriately and rigorously? 

Reviewer #1: Yes

Reviewer #2: Yes

3. Have the authors made all data underlying the findings in their manuscript fully available?

Reviewer #1: Yes

Reviewer #2: No

4. Is the manuscript presented in an intelligible fashion and written in standard English?

Reviewer #1: Yes

Reviewer #2: Yes

5. Review Comments to the Author

Reviewer #1: Dear authors,

Overall, this was a good paper and nice to see more research linking facility and household survey data. I have some comments for further clarifications and some suggestions for your analysis. I hope my suggestions will further improve your paper.

Thank you.

1. Quality of care index outcome – The variables listed in Appendix 1 are mainly related to counseling or what was told to the client by the provider. Are there some other components of QoC that can also be added? What about ensured privacy (private room during consultation), or following medical guidelines (washed hands, sanitized equipment) that may be observed during the survey.

2. How were the two datasets linked? The NFHS data have displacement procedures to protect the privacy of individuals. The facility survey is also not a census but a sample. We cannot know which facility the women went to, but the assumption is she went to a nearby facility in her district. However, she could have went to facility in another district if there was no facility in her district that had fertilization procedures or was of low quality. The two data sources are three years apart however and women may have moved in this time. These methodological issues should be explained more clearly and discussed as limitations to the analysis.

3. Other background variables that would have been important to include, are women’s number of living children, and perhaps information about the provider if available such education level or number of years of experience, or type of provider. Perhaps also if there is information on whether this was the women’s first visit to the facility. This may influence her judgment on the QoC of the facility.

4. You are modeling the log odds of good to high QoC correct? In the description of your equation you said good QoC.

5. In the methods, you need to describe how the multilevel models were fit in Stata. To account for the multi-stage survey design, the svy command should be used and for multi-level models a weight is required for each level. Please describe which weights were used and how you accounted for the survey design. If this was not done, then you will underestimate your SEs. Was ArcGIS only used to create the maps? This was not clear.

6. Since you describe low QoC in Table 1 and Figures 1 and 2, I believe it makes more sense to model low QoC in the regressions and not good to high QoC. This also makes sense since QoC is relatively high and there is little variability. Also, from a programmatic sense you would want to identify where the low QoC is for interventions.

7. The paper is not consistent in how you refer to the outcome. Sometimes you say good QoC, sometimes excellent, sometimes very high. Please be consistent so it is not confusing.

8. Interpretation of odds ratios are not exactly correct. Example 8% less likely. These are not likelihoods. Please say 8% lower odds for instance. Also, an AOR of 0.926 rounds to 7% lower odds and not 8%. I suggest rounding the AORs to 1 or 2 decimal places and checking the numbers used in the text.

9. Since most of the variables used to construct the QoC outcome are about the providers counseling or what they told the client, some literature on provider counseling or provider-client interactions would be useful both in the introduction and discussion.

Reviewer #2: This paper is very well written and the analysis has been done meticulously. It highlights an important issue of the quality of care of sterilization services in India. However, I believe that there are some places that this article could modify. First of all, there are a few corrections in the language of the article. For example, in the fourth paragraph of the discussion, in the fourth line, the word "centers" is incorrectly typed as "Canters". There are a few other grammatical mistakes for which the author should re check the article manuscript. Next, it would be of use if the authors could specifically suggest in the article in the discussion part about what could be helpful to improve the quality of care of sterilization services at each tier of the public health system on both the provider's side and the beneficiary's side at all three structure-process-outcome levels.

6. PLOS authors have the option to publish the peer review history of their article (what does this mean?). If published, this will include your full peer review and any attached files.

Reviewer #1: **Yes: **Shireen Assaf

Reviewer #2: **Yes: **Yash Alok

---

## [Author Response · Author response to Decision Letter 0]

2 Oct 2020

Journal Requirements:

Answer: Thank you for noting this. We have updated the manuscript according to the Journal guidelines.

Answer: The NFHS 2015–16 and DLHS 2012–13 received ethical clearance from the Ethical Review Board of the International Institute for Population Science, which conducted these surveys. The analysis presented in this study is based on a secondary analysis of existing survey data, available in the public domain without any identifying information. The survey personnel obtained informed consent from each respondent before the interview and made their best effort to maintain privacy.

Answer: Data of NFHS are available in the public domain without any identifiers in the following website—https://dhsprogram.com/data/available-datasets.cfm.

Data of DLHS are available in the public domain in the following website—http://rchiips.org/DLHS-4.html.

Answer: Data are available in public domain.

4. Please ensure that you refer to Figure 1 in your text as, if accepted, production will need this reference to link the reader to the figure.

Answer: Thank you for noting this. As you suggested, we have cited the Figure.

5. We note that Figures 2 and 3 in your submission contain map images which may be copyrighted. All PLOS content is published under the Creative Commons Attribution License (CC BY 4.0), which means that the manuscript, images, and Supporting Information files will be freely available online, and any third party is permitted to access, download, copy, distribute, and use these materials in any way, even commercially, with proper attribution. For these reasons, we cannot publish previously copyrighted maps or satellite images created using proprietary data, such as Google software (Google Maps, Street View, and Earth). For more information, see our copyright guidelines: http://journals.plos.org/plosone/s/licenses-and-copyright.

Answer: The authors created those original maps in Figures 2 and 3 from NFHS-4 and DLHS-4 data, respectively. Therefore, authors currently hold copyright.

The shapefiles used to create Fig 2, and Fig 3 were downloaded from the DHS program spatial data repository, which can be downloaded without any restriction along with the NFHS data. For more details visit DHS website to obtain the GPS datasets.

https://dhsprogram.com/data/available-datasets.cfm

One of the authors (VJ) has modified the boundaries of Jammu & Kashmir region in the shapefile using GIS mapping software so that the map displays in accordance to the official map recognized by the government of India. The author has the modified shapefile.

 5. 1. You may seek permission from the original copyright holder of Figures 2 and 3 to publish the content specifically under the CC BY 4.0 license. 

 5.2. If you are unable to obtain permission from the original copyright holder to publish these figures under the CC BY 4.0 license or if the copyright holder’s requirements are incompatible with the CC BY 4.0 license, please either i) remove the figure or ii) supply a replacement figure that complies with the CC BY 4.0 license. Please check copyright information on all replacement figures and update the figure caption with source information. If applicable, please specify in the figure caption text when a figure is similar but not identical to the original image and is therefore for illustrative purposes only.

Reviewers' comments:

Reviewer's Responses to Questions

Comments to the Author

1. Is the manuscript technically sound, and do the data support the conclusions?

Reviewer #1: Partly

Reviewer #2: Yes

2. Has the statistical analysis been performed appropriately and rigorously?

Reviewer #1: Yes

Reviewer #2: Yes

3. Have the authors made all data underlying the findings in their manuscript fully available?

Reviewer #1: Yes

Reviewer #2: No

4. Is the manuscript presented in an intelligible fashion and written in standard English?

Reviewer #1: Yes

Reviewer #2: Yes

5. Review Comments to the Author

Reviewer #1: Dear authors,

Overall, this was a good paper and nice to see more research linking facility and household survey data. I have some comments for further clarifications and some suggestions for your analysis. I hope my suggestions will further improve your paper.

Thank you.

1. Quality of care index outcome – The variables listed in Appendix 1 are mainly related to counseling or what was told to the client by the provider. Are there some other components of QoC that can also be added? What about ensured privacy (private room during consultation), or following medical guidelines (washed hands, sanitized equipment) that may be observed during the survey.

Answer: Authors agree with the reviewer that other QoC indicators such as privacy or following the standard of protocols are important. However, NFHS does not provide those data. This study used only those QoC indicators which are available in the NFHS data, and those five variables were included in the analyses.

2. How were the two datasets linked? The NFHS data have displacement procedures to protect the privacy of individuals. The facility survey is also not a census but a sample. We cannot know which facility the women went to, but the assumption is she went to a nearby facility in her district. However, she could have went to facility in another district if there was no facility in her district that had fertilization procedures or was of low quality. The two data sources are three years apart however and women may have moved in this time. These methodological issues should be explained more clearly and discussed as limitations to the analysis.

Answer: We have mentioned, in the method section, that we calculated facility readiness indicators at the district level using DLHS facility data of 2012–13. We only considered the data of District Hospitals and Community Health Centers (CHC) because all DH and CHCs within the district were covered in the DLHS facility survey, and no sampling procedure is involved. We assumed that the facility readiness scores at the district level will provide a proxy estimation of the structural aspect of QoC readiness in the district.

The QoC process-outcome indicator was calculated at the individual level using NFHS 2015–16 data. For the multilevel analysis, we merged the individual-level dataset and district-level variables following ‘many to one’ procedure, so in the merged dataset all women from a district had the same score for facility readiness. 

The DLHS facility survey 2012–13 does not provide data from all Indian districts and data from 560 districts are available. Therefore, we have only used the NFHS women data form only those 560 districts out of a total of 640 Indian districts in the dataset.

Yes, we do agree with the reviewer that two data sources are three years apart observations that the women could have gone to a facility in another district. While merging NFHS 2015–16 and DLHS 2012–13 datasets, we assumed that there is not much structural variation in infrastructure in the two survey periods. Due to this assumption, the changes in structural factors during this period are unaddressed. On the other hand, many of the women, who adopted sterilization, underwent the procedure well before the survey year so even if we have the data from the same year the direct one-to-one matching of facility readiness and QoC process-outcome is not possible. 

In this study, DLHS 2012–13 data for ‘structure score’ serves as the proxy measurement for the system and facility readiness which is a precondition for having quality service. We considered all these issues as the limitations of the study, and we have addressed in the second last paragraph of the discussion section.

3. Other background variables that would have been important to include, are women’s number of living children, and perhaps information about the provider if available such education level or number of years of experience, or type of provider. Perhaps also if there is information on whether this was the women’s first visit to the facility. This may influence her judgment on the QoC of the facility.

Answer: The authors agree to the reviewer that those are some important variables to consider. However, the variables about a woman’s visit to the facility or the education, years of experience, or type of provider who was conducted the sterilization procedure are not available in the NFHS dataset.

In this analysis, the dependent variable is the QoC of the sterilization and the existing literature on QoC did not show that the number of children of the woman could be a determining factor for the quality of care a woman receives for sterilization procedure. Therefore, we did not consider that as an independent variable.

4. You are modeling the log odds of good to high QoC correct? In the description of your equation you said good QoC.

Answer: In revised manuscript receipt of ‘low’ QoC has been considered for modelling.

5. In the methods, you need to describe how the multilevel models were fit in Stata. To account for the multi-stage survey design, the svy command should be used and for multi-level models a weight is required for each level. Please describe which weights were used and how you accounted for the survey design. If this was not done, then you will underestimate your SEs. Was ArcGIS only used to create the maps? This was not clear.

Answer: We agree that to account for the multi-stage survey design the svy command should be used. The survey final weights are not allowed with multilevel models. In NFHS the sample weights were calculated for the cluster level (PSUs). Since the multilevel models require each stage-level weight variable using the stage’s corresponding weight such that the weights should be constant within the groups defined. In this study, the respondents comprise the first level, and the districts comprise the second level. As we didn’t consider cluster (or PSU) as a level and weights were not constant within the district, we didn’t use the sampling weights in the regression analysis. Yes, ArcGIS software was used to create the maps.

6. Since you describe low QoC in Table 1 and Figures 1 and 2, I believe it makes more sense to model low QoC in the regressions and not good to high QoC. This also makes sense since QoC is relatively high and there is little variability. Also, from a programmatic sense you would want to identify where the low QoC is for interventions.

Answer: Thank you for your comment. We have changed the coding of the dependent variable and recreated the table and text.

7. The paper is not consistent in how you refer to the outcome. Sometimes you say good QoC, sometimes excellent, sometimes very high. Please be consistent so it is not confusing.

Answer: Thank you for noting this. We have revised the manuscript addressing your suggestion.

8. Interpretation of odds ratios are not exactly correct. Example 8% less likely. These are not likelihoods. Please say 8% lower odds for instance. Also, an AOR of 0.926 rounds to 7% lower odds and not 8%. I suggest rounding the AORs to 1 or 2 decimal places and checking the numbers used in the text.

Answer: Thank you for noting this. We have revised the manuscript as per journal guideline.

9. Since most of the variables used to construct the QoC outcome are about the providers counseling or what they told the client, some literature on provider counseling or provider-client interactions would be useful both in the introduction and discussion.

Answer: Thank you for noting this. As you suggested, we have updated the manuscript.

Reviewer #2: This paper is very well written and the analysis has been done meticulously. It highlights an important issue of the quality of care of sterilization services in India. However, I believe that there are some places that this article could modify. First of all, there are a few corrections in the language of the article. For example, in the fourth paragraph of the discussion, in the fourth line, the word "centers" is incorrectly typed as "Canters". There are a few other grammatical mistakes for which the author should re check the article manuscript. 

Next, it would be of use if the authors could specifically suggest in the article in the discussion part about what could be helpful to improve the quality of care of sterilization services at each tier of the public health system on both the provider's side and the beneficiary's side at all three structure-process-outcome levels.

Answer: We appreciate your insightful suggestions. As you suggested, we have thoroughly revised the manuscript addressing your suggestion.

6. PLOS authors have the option to publish the peer review history of their article (what does this mean?). If published, this will include your full peer review and any attached files.

Do you want your identity to be public for this peer review? For information about this choice, including consent withdrawal, please see our Privacy Policy.

Reviewer #1: Yes: Shireen Assaf

Reviewer #2: Yes: Yash Alok

Answer: We have checked all our figures using PACE digital diagnostic tool before uploading.

---

## [Editor Report · Decision Letter 1]

16 Oct 2020

Quality of care in sterilization services at the public health facilities in India: A multilevel analysis

PONE-D-20-22065R1

Dear Dr. Mozumdar,

We’re pleased to inform you that your manuscript has been judged scientifically suitable for publication and will be formally accepted for publication once it meets all outstanding technical requirements.

Kind regards,

Kannan Navaneetham, PhD

Academic Editor

PLOS ONE
---

## [Editor Report · Acceptance letter]

22 Oct 2020

PONE-D-20-22065R1 

Quality of care in sterilization services at the public health facilities in India: A multilevel analysis 

Dear Dr. Mozumdar:

I'm pleased to inform you that your manuscript has been deemed suitable for publication in PLOS ONE. Congratulations! Your manuscript is now with our production department. 

Kind regards, 

on behalf of

Professor Kannan Navaneetham 

Academic Editor

PLOS ONE